# Potassium Iodide Doping for Vacancy Substitution and Dangling Bond Repair in InP Core-Shell Quantum Dots

**DOI:** 10.3390/nano14121055

**Published:** 2024-06-19

**Authors:** Ji-Eun Lee, Chang-Jin Lee, Seung-Jae Lee, Ui-Hyun Jeong, Jea-Gun Park

**Affiliations:** 1Department of Information Display Engineering, Hanyang University, Seoul 04763, Republic of Korea; lge7825@hanyang.ac.kr; 2Department of Electronic Engineering, Hanyang University, Seoul 04763, Republic of Korea; changjin0479@hanyang.ac.kr (C.-J.L.); sseungjae90@hanyang.ac.kr (S.-J.L.); uiuh1245@hanyang.ac.kr (U.-H.J.); 3Samsung Electronics, 130 Samsung-ro, Suwon 16678, Republic of Korea

**Keywords:** quantum dots, indium phosphide, vacancy, dangling bond, doping, alkali metal, potassium iodide, quantum dots functional color filter, color representation

## Abstract

This work highlights the novel approach of incorporating potassium iodide (KI) doping during the synthesis of In_0.53_P_0.47_ core quantum dots (QDs) to significantly reduce the concentration of vacancies (i.e., In vacancies; *V*_In_^−^) within the bulk of the core QD and inhibit the formation of InPO_x_ at the core QD–Zn_0.6_Se_0.4_ shell interfaces. The photoluminescence quantum yield (PLQY) of ~97% and full width at half maximum (FWHM) of ~40 nm were achieved for In_0.53_P_0.47_/Zn_0.6_Se_0.4_/Zn_0.6_Se_0.1_S_0.3_/Zn_0.5_S_0.5_ core/multi-shell QDs emitting red light, which is essential for a quantum-dot organic light-emitting diode (QD-OLED) without red, green, and blue crosstalk. KI doping eliminated *V*_In_^−^ in the core QD bulk by forming K^+^-*V*_In_^−^ substitutes and effectively inhibited the formation of InPO_4_(H_2_O)_2_ at the core QD–Zn_0.6_Se_0.4_ shell interface through the passivation of phosphorus (P)-dangling bonds by P-I bonds. The elimination of vacancies in the core QD bulk was evidenced by the decreased relative intensity of non-radiative unpaired electrons, measured by electron spin resonance (ESR). Additionally, the inhibition of InPO_4_(H_2_O)_2_ formation at the core QD and shell interface was confirmed by the absence of the {210} X-ray diffraction (XRD) peak intensity for the core/multi-shell QDs. By finely tuning the doping concentration, the optimal level was achieved, ensuring maximum K-*V*_In_^−^ substitution, minimal K^+^ and I^−^ interstitials, and maximum P-dangling bond passivation. This resulted in the smallest core QD diameter distribution and maximized optical properties. Consequently, the maximum PLQY (~97%) and minimum FWHM (~40 nm) were observed at 3% KI doping. Furthermore, the color gamut of a QD-OLED display using R-, G-, and B-QD functional color filters (i.e., ~131.1%@NTSC and ~98.2@Rec.2020) provided a nearly perfect color representation, where red-light-emitting KI-doped QDs were applied.

## 1. Introduction

Among various luminescent materials, quantum dots (QDs) have been intensively researched owing to their simple wavelength tunability, high efficiency, high stability, and high color purity [1,2,3,4,5,6,7,8,9,10]. Consequently, in recent decades, research on the application of QDs in fields such as solar energy [11,12,13,14], bioimaging [15,16], and displays [10,17,18,19,20,21] has been vigorously pursued. In particular, luminescent materials for display applications, such as quantum-dot organic light-emitting diodes (QD-OLED) and quantum-dot light-emitting diodes (QLED), are required to achieve a photoluminescence quantum yield (PLQY) of >95% and a full width at half maximum (FWHM) of <30 nm [1,10,22,23,24]. In addition, recent studies have explored the integration of QDs with GaN-based LEDs for full-color displays. For instance, Zhou et al. [25] investigated high-efficiency GaN-based green LEDs utilizing InGaN quantum wells with varying indium content, demonstrating improvements in light output power and reduction in efficiency droop. Fan et al. [26] examined the development of efficient full-spectrum WLEDs through monolithic integration of III-nitride quantum structures via bandgap engineering, achieving notable advancements in color rendering and luminous efficacy. These studies indicate ongoing efforts to enhance display technologies by combining QDs with GaN-based LEDs. For instance, quantum dots based on II–VI materials, such as cadmium selenide (CdSe) QDs, exhibit excellent optical properties, including a high PLQY (>95%), high stability, and a narrow FWHM (>25 nm) [2,27,28,29,30]. However, owing to environmental concerns, the use of Cd in display applications has been restricted according to the specifications of the European Restriction of Hazardous Substances Directive (RoHS) [31,32,33,34]. In contrast, luminescent nanomaterials for display applications based on III-V materials, such as indium phosphide (InP), comply with RoHS standards. In particular, InP-based QDs, which are III-V materials with a bulk bandgap energy of approximately 1.35 eV [35], allow for the adjustment of their energy from near-blue (~2.5 eV) to near-infrared (~1.7 eV) through control of the core diameter [36,37,38,39]. Although the environmentally friendly InP-based QDs have been actively researched, their high lattice covalency presents a challenge for achieving a high PLQY of >95%, leading to an inherently low PLQY for InP-based QDs [3,40,41,42,43,44,45,46]. Lattice covalency, which can be characterized by the Phillips’ ionicity, represents the quantified value of the type of chemical bonding between ionic bonding (i.e., higher Phillips’ ionicity) and covalent bonding (i.e., lower Phillips’ ionicity) [47,48]. The Phillips’ ionicity of InP-based QDs is relatively low due to the covalent bonding between In and P atoms, resulting in a higher lattice covalency compared with CdSe-based QDs. The Phillips’ ionicity values for InP and CdSe are 0.421 and 0.699, respectively [49]. A high lattice covalency of InP-based QDs inherently requires a high growth temperature (i.e., >300 °C) and a reactive P^3−^ precursor (i.e., tris(trimethylsilyl) phosphine; (TMS)_3_P) [10,47,50,51,52,53,54,55]. InP-based QDs grown at a high temperature exhibit internal defects such as vacancies in the InP core QDs (i.e., In^−^ vacancies (*V*_In_^−^) and P^+^ vacancies (*V*_P_^+^) in the core QDs) during nucleation and growth [56,57,58], resulting in a low PLQY and wide FWHM [53,59,60,61,62]. Recent advancements in doping methods for InP QDs have shown promising improvements in their optical properties. For instance, various metal impurities such as Cu, Ag, and Au have been successfully introduced into semiconductor nanocrystals, demonstrating control over the bandgap and Fermi energy, which significantly influences the photoluminescence (PL) and electronic properties of the QDs [63,64]. The introduction of dopants during the synthesis process or post-synthesis treatment has been explored to enhance the PLQY and stability of QDs by minimizing the non-radiative recombination sites through improved crystallinity and surface passivation [64]. Additionally, doping strategies involving surface ligand exchange and electrochemical doping have been reported to tailor the electronic and optical properties of QDs, thus optimizing their performance in various applications including display technologies and bioimaging [63,64]. In addition, the surface of the InP-core QDs can be readily oxidized, resulting in InPO_x_ [4,50,56,65,66,67,68]. Thus, optimal hydrofluoric acid (HF) treatment is generally introduced prior to shell growth [10,69,70,71,72,73,74,75,76,77]. Moreover, halide ion diffusion and passivation after InP/ZnSeS/ZnS core/shell growth were applied to inhibit the oxidation of the InP core surface, reduce the number of interface defects between the InP core and ZnSeS shell and between the ZnS outer shell, and diminish vacancies such as *V*_In_^−^ and *V*_P_^+^ [78]. However, the action mechanism has not been elucidated.

In our study, doping with a metal halide (i.e., potassium iodide; KI) was precisely designed during In_x_P_1−x_ core synthesis, followed by optimal HF treatment and the multi-shell growth of a Zn_0.6_Se_0.4_/Zn_0.6_Se_0.1_S_0.3_/Zn_0.5_S_0.5_ nanolayer, as shown in Figure 1a. Note that the In_x_P_1−x_ core QDs and Zn_0.6_Se_0.4_/Zn_0.6_Se_0.1_S_0.3_/Zn_0.5_S_0.5_ multi-shells were precisely designed to maximize the PLQY (i.e., ~97%) and minimize the FWHM (i.e., ~40 nm) under 622 nm red-light emission. As shown in Figure 1a, the In_x_P_1−x_ core doped with an optimal concentration of KI had a diameter of 4.1 nm ± 0.5 nm. Following the shell growth process described in the next Section 2.2, Zn_0.6_Se_0.4_, Zn_0.6_Se_0.1_S_0.3_, and Zn_0.5_S_0.5_ multi-shell layers were grown, resulting in a final core-shell QD structure with dimensions of approximately 7.5 nm. Moreover, Figure 1b,c schematically represent the concept of improving optical properties through our designed doping process, illustrating the aim to obtain structure; Figure 1b shows the In_x_P_1−x_ core with internal vacancies and their substitution by KI doping, while Figure 1c demonstrates the passivation of surface oxidation, specifically inhibiting the formation of oxidized InPO_x_ on the core QD surface through KI doping. In particular, among metal halides, KI was selected to minimize the ionic-radius mismatch between vacancies (i.e., *V*_In_^−^ or *V*_P_^+^) and metal halides (i.e., K^+^ or I^−^), minimizing the degree of non-radiative recombination, where the ionic radii of In^3+^, P^3−^, K^+^, I^−^, *V*_In_^−^, and *V*_P_^+^ were 80, 212, 138, 220, 155, and 100 pm, respectively [79,80], as shown in Figure 1b.

In addition, the effect of K^+^ or I^−^ doping on the passivation efficiency of *V*_In_^−^ and *V*_P_^+^ in only the In_x_P_1−x_ core QDs was investigated as a function of the KI doping concentration. The core QD average diameter and diameter distribution were investigated using high-resolution transmission electron microscopy (HR-TEM), and the relative vacancy concentration was investigated using electron spin resonance (ESR). Moreover, to examine the effect of the KI doping concentration on the photoelectric performance enhancement, the photo-optical properties of the In_x_P_1−x_/Zn_0.6_Se_0.4_/Zn_0.6_Se_0.1_S_0.3_/Zn_0.5_S_0.5_ core/multi-shell QDs were estimated as a function of the KI doping concentration by measuring the emission wavelength, PLQY, FWHM, and exciton lifetime using time-resolved photoluminescence (TRPL) spectra. Furthermore, to determine the mechanism whereby KI doping during core synthesis significantly enhanced the photo-optical performance, the dependence of the crystalline properties of the core/multi-shell QDs on the KI doping concentration was precisely characterized using X-ray diffraction (XRD). Finally, KI-doped InP-based red-light-emitting QDs were applied in a hybrid-display application that combined quantum-dot functional color filters (QDCF) and a blue OLED backlight unit (BLU) [81,82,83,84]. The color gamut performance was evaluated by comparing the CIE1931 x,y color coordinates with the Rec.2020 color standard and the National Television System Committee (NTSC) color standard [85,86,87]. The NTSC standard, established in 1953, defines a color gamut based on the RGB color model for CRT displays, while Rec.2020, introduced by the International Telecommunication Union (ITU) in 2012, encompasses a significantly larger range of colors for Ultra-High-Definition (UHD) television. The absence of crosstalk between the primary colors was confirmed by measuring the polarized R-, G-, and B-light photoluminescence (PL) spectra.

## 2. Materials and Methods

### 2.1. Materials

For the core and shell materials of QDs, trioctylphosphine (TOP, 97%, Sigma-Aldrich, St. Louis, MO, USA), 1-octadecene (ODE, technical 90%, Sigma-Aldrich, St. Louis, MO, USA), indium(III) acetate (In(OAc)3, 99.99%, Sigma-Aldrich, St. Louis, MO, USA), zinc stearate (Zn(st)2, technical grade, Sigma-Aldrich, St. Louis, MO, USA), palmitic acid (PA, >99%, Sigma-Aldrich, St. Louis, MO, USA), potassium iodide (KI, 99%, Sigma-Aldrich, St. Louis, MO, USA), selenium (Se, 99.99% powder 100 mesh, Sigma-Aldrich, St. Louis, MO, USA), sulfur (trace metal basis 99.998%, Sigma-Aldrich, St. Louis, MO, USA), 1-dodecanethiol (DDT, >98%, Sigma-Aldrich, St. Louis, MO, USA), hydrofluoric acid (HF,48%, Sigma-Aldrich, St. Louis, MO, USA), Toluene (>99.9%, Daejung, Siheung, Republic of Korea), hexane (>98.5%, Daejung, Siheung, Republic of Korea), acetone (>99.8%, Daejung, Siheung, Republic of Korea), ethanol (>99.5%, Daejung, Siheung, Republic of Korea), n-octane (>97%, Daejung, Siheung, Republic of Korea), and Tris(trimethylsilyl) phosphine ((TMS)3P, 99.5%, SK Chemical, Seongnam, Republic of Korea) were employed. In addition, for the materials of red-, green-, and blue-color filters, DCR-TR711R, DCR-TR711G, and DCR-TR711B (Dongjin Semichem Co., Seoul, Republic of Korea) were used, respectively.

### 2.2. Synthesis of Core/Multi-Shell QDs

#### 2.2.1. Preparation of Stock Solutions

For the synthesis of red-light-emitting QDs, a 0.316 M Zn(st)_2_ precursor was prepared by dissolving 4.74 mmol of Zn(st)_2_ in 15 mL of ODE, and the mixed solution was degassed and heated using the same conditions. A 0.2 M solution of (TMS)_3_P was prepared in a N_2_-filled glovebox by combining 2 mmol of (TMS)_3_P with 10 mL of TOP. A 1.79 M Se-TOP mixed solution was prepared in an N_2_-filled glovebox by dissolving 17.9 mmol of Se powder in 10 mL of TOP. A 0.1M HF-acetone mixed solution was prepared by dissolving 1.4 mmol of HF in 14 mL of acetone. After that, all mixed solutions were degassed at 200 °C for 30 min in an N_2_-filled glovebox.

#### 2.2.2. Synthesis of KI-Doped Red-Light-Emitting In_x_P_1−x_ Core QDs

For the synthesis of KI-doped In_0.53_P_0.47_ core QDs, 0.65 mmol of In(OAc)_3_, 0.002 mmol of KI, and 1.95 mmol of PA were loaded into a 100 mL, 3-neck flask with 14 mL of ODE at RT. The mixed solution in flask was heated up to 150 °C with stirring and degassed under a vacuum of 100 mTorr for 60 min. Afterward, the degassed mixed solution in flask was heated up to 320 °C to obtain a colorless transparent In(PA)_3_ solution. After that, the mixed solution in syringe of 1.63 mL of 0.2 mM (TMS)_3_P precursor was rapidly injected into the flask at 320 °C. Next, the KI-doped In_x_P_1−x_ core was grown for 10 min at the same temperature. And then, to obtain red-light-emitting KI-doped In_x_P_1−x_ core QDs, the mixed solution in flask was cooled down to RT and then centrifuged twice with acetone to eliminate impurities generated from unreacted precursors and byproducts. Finally, the precipitated QDs were redispersed in 5 mL of toluene.

#### 2.2.3. Synthesis of Red-Light-Emitting KI-Doped In_x_P_1−x_/Zn_0.6_Se_0.4_/Zn_0.6_Se_0.1_S_0.3_/Zn_0.5_S_0.5_ Core/Multi-Shell QDs

To grow a Zn_0.6_Se_0.4_ shell on the synthesized KI-doped In_0.53_P_0.47_ core QDs, a reaction flask was prepared using a 100 mL, 3-neck flask by mixing 2.1 mL of 0.316 M Zn(st)_2_ with 15 mL of ODE. The mixed solution in flask was stirred and heated up to 150 °C, then degassed under a pressure of 100 mTorr for 60 min. Subsequently, KI-doped In_x_P_1−x_ core QDs in 5 mL of toluene were injected into the flask at 150 °C and treated with 1.4mL of HF-acetone under a N_2_ flow. The mixed solution in flask was heated up to 210 °C, and then 0.20 mL of 1.79 M Se-TOP precursor was injected and kept at 210 °C for 10 min. After that, the mixed solution in syringe of 2.5 mL of 0.316 M Zn(st)_2_ precursor was injected dropwise, and the mixed solution in the flask was kept at 240 °C for 10 min. Next, the mixed solution in a syringe of 0.16 mL of 1.79 M Se-TOP precursor was injected, and the mixed solution in the flask was kept at 270 °C for 10 min. Afterward, the mixed solution in the syringe of 2.9 mL of 0.316 M Zn(st)_2_ precursor was injected dropwise, and the mixed solution in the flask was kept at 300 °C for 10 min. And then, the mixed solution in the syringe of 0.16 mL of 1.79 M Se-TOP precursor was injected, the mixed solution in the flask was kept at 330 °C for 10 min. In addition, for the passivation of Zn_0.6_Se_0.4_/Zn_0.6_Se_0.1_S_0.3_ multi-shell, the mixed solution in flask was cooled down to 260 °C, and the mixed solution in syringe of 1.4 mL of 0.316 M Zn(st)_2_ precursor was injected dropwise at this temperature and kept for 10 min. After that, the mixed solution in the syringe of 0.45 mL of Se_0.3_S_0.7_-TOP precursor was dropped one drop at a time, the mixed solution in the flask was kept at 300 °C for 20 min. Next, the mixed solution in the syringe of 1.4 mL of 0.316 M Zn(st)_2_ precursor was injected dropwise, the mixed solution in the flask was kept at 320 °C for 20 min. And then, the mixed solution in the syringe of 0.45 mL of Se_0.15_S_0.85_-TOP precursor was injected dropwise, the mixed solution in the flask was kept at 320 °C for 40 min. Furthermore, for the passivation of Zn_0.5_S_0.5_ outer-shell, the mixed solution in the flask of the KI-doped In_x_P_1−x_/Zn_0.6_Se_0.4_/Zn_0.6_Se_0.1_S_0.3_ core/multi-shell was cooled down to 230 °C, and 8.8 mL of 0.27 M Zn(OA)_2_ precursor was injected and kept at 230 °C for 30 min. And then, 0.96 mL of the DDT precursor was injected into the flask, and the temperature was kept. After 30 min, to obtain red-light-emitting KI-doped In_0.53_P_0.47_ /Zn_0.6_Se_0.4_/Zn_0.6_Se_0.1_S_0.3_/Zn_0.5_S_0.5_ core/multi-shell QDs, the flask was cooled down to RT and then centrifuged three times with acetone and ethanol to eliminate impurities generated from unreacted precursors and byproducts. Finally, the precipitated QDs were redispersed in hexane and stored at RT for the application in QD-functional CF-OLED hybrid display applications. A schematic synthesis process is presented in Appendix A.

### 2.3. Characterizations

The morphology, size distribution, and core/shell structure of KI-doped In_x_P_1−x_/Zn_0.6_Se_0.4_/Zn_0.6_Se_0.1_S_0.3_/Zn_0.5_S_0.5_ core/multi-shell QDs were characterized using transmission electron microscopy (TEM, JEM-2100F from JEOL, Tokyo, Japan) at an acceleration voltage of 200 kV. The actual KI doping amounts were quantified using ICP-AES (model: OPTIMA 8300, Perkin-Elmer, Waltham, MA, USA) to determine the K^+^/In^3+^ mole ratio. Additionally, defects in the In_x_P_1−x_ core QDs were observed using electron spin resonance (ESR) spectroscopy (model: JES-FA200, maker: JEOL, Tokyo, Japan) with a central field of 3505 milli Tesla (mT). Optical properties were further characterized by obtaining ultraviolet-visible absorption spectra (model: Cary 5000, maker: Agilent, Santa Clara, CA, USA) and photoluminescence (PL) spectra (model: DM-700i, maker: Dongwoo Optron, Gwangju, Republic of Korea) using a He-Cd laser source at an excitation wavelength of 325 nm. The full width at half maximum (FWHM) and absolute photoluminescence quantum yield (PLQY) were extracted using a QE-2100 (from Otsuka Electronics, Hirakata, Osaka, Japan) with a 150 W xenon lamp light source at an excitation wavelength of 450 nm. The QE-2100 system provides high-resolution measurements, capable of measuring peak wavelengths and FWHM with a precision specified in nm based on the model used. For clarity in our manuscript, the peak wavelength and FWHM values were rounded to the nearest nm, and internal quantum efficiency (IQE) values were converted to percentages and represented as whole numbers. Moreover, PLQY was measured using an integrating sphere setup, which calculates the ratio of the number of photons emitted to the number of photons absorbed, ensuring accurate absolute quantum efficiency measurements. Measurements were performed using a 150 mm integrating hemisphere in a liquid sample state, with fluorescein solution at 493 nm excitation wavelength as the reference. The internal quantum efficiency (yield) of fluorescein was calculated as 0.903 (concentration: 6.43 × 10^−6^ mol∙L^−1^), matching the literature value [88]. The reference measurement was followed by the sample measurement, and a correction for re-excitation was applied to determine the final internal quantum efficiency.
$$PLQY=\frac{Number\;of\;photons\;emitted}{Number\;of\;photons\;absorbed}\times 100\%$$

In particular, the exciton lifetime of the QDs was determined by using time-correlated single-photon counting measurements with single-photon avalanche diodes (PDL Series, from PicoQuant, Berlin, Germany) and a HydraHarp 400 multichannel (from PicoQuant, Berlin, Germany) picosecond event timer module. Furthermore, the crystallinity and crystal structure of the core/multi-shell QDs were characterized by X-ray diffraction (XRD).

## 3. Results and Discussion

### 3.1. Dependence of Defect Passivation Efficiency on KI Dopant Concentration for KI-Doped In_x_P_1−x_ Core QDs

To estimate the effect of KI doping on the passivation of vacancies (i.e., *V*_In_^−^ or *V*_P_^+^) and surface defects (i.e., oxidation of P-dangling bonds), only KI-doped In_x_P_1−x_ core QDs were synthesized by varying the KI doping concentration from 1% to 7% ([KI/Indium precursor] molar ratio). The actual KI doping concentrations were confirmed by ICP-AES analysis, showing a linear increase in the K^+^/In^3+^ molar ratio from 0 to 0.02 as the KI doping concentration increased from 0% to 7%, as detailed in the Appendix A. The dependence of the crystalline properties of the KI-doped core QDs on the KI doping concentration was examined using HR-TEM, as shown in Figure 2. The undoped core QDs were well crystallized, with a zincblende structure having a distance of 3.42 Å between {111}. The average QD diameter and diameter distribution were 3.7 and ±1.0 nm, as shown in Figure 2a. Note that the core average diameter and its deviation were determined by measuring the diameters of well-dispersed core QD particles within a defined window size of 100 nm × 100 nm. This window size was used to ensure a representative sample of the particle population. By measuring a sufficient number of particles (at least 200 particles) within this window, we obtained the average diameter and the standard deviation, reflecting the size distribution. The KI doping in the core QDs significantly increased the average core QD diameter from 3.7 to 4.6 nm, and all the KI-doped core QDs were well crystallized with a zincblende structure, having a distance of 3.38–3.42 Å between {111}, as shown in Figure 2a–f. This result would indicate that the dissociated K^+^ and I^−^ not only substitute for *V*_In_^−^ and *V*_P_^+^ but also produce interstitial K^+^ and I^−^ within the In_x_P_1−x_ core QDs. The ionic radii of In^3+^, P^3−^, K^+^, I^−^, *V*_In_^−^, and *V*_P_^+^ were 80, 212, 138, 220, 155, and 100 pm, respectively. When the KI doping from 0% to 3%, the diameter distribution of the core QDs narrowed from ±1.0 to ±0.5 nm, as shown in Figure 2a–c,f. Then, it considerably widened from ±0.5 to ±1.4 nm when the KI doping concentration increased from 3% to 7%, as shown in Figure 2c–f. The minimum diameter distribution of the In_x_P_1−x_ core QDs was observed at a specific KI doping concertation (i.e., 3% KI). The results from increasing the KI doping concentration from 0% to 3% indicated that substituting K^+^ and I^−^ for *V*_In_^−^ and *V*_P_^+^ enhanced the uniformity of the core QD diameter distribution, as a reduction in *V*_In_^−^ and *V*_P_^+^ leads to more homogeneous QD growth. However, the increase in the KI doping concentration further deteriorated the uniformity of the QD diameter distribution because the generation of K^+^ and I^−^ interstitials, instead of substitutes, resulted in inhomogeneous In_x_P_1−x_ core QDs.

To determine why the minimum diameter distribution of the In_x_P_1−x_ core QDs was observed at a specific KI doping concentration (i.e., 3% KI) via K^+^ and I^−^ substitutes, the dependence of the integrated ESR signal on the KI doping concentration was investigated for only KI-doped core QDs. Note that electromagnetic waves within the GHz (microwave) range were utilized in ESR spectroscopy. The signals arose from the interaction between the unpaired electrons in the sample and the applied external magnetic field, which was attributed to the Zeeman effect [89,90,91,92,93,94]. During the synthesis of the In_x_P_1−x_ core QDs, the internal defects such as *V*_In_^−^ and *V*_P_^+^ or K^+^ and I^−^ interstitials in the core bulk, as well as In- and P-dangling bonds on the core QD surface, can generate observable electron spin states in ESR analysis. These spin states, which are associated with the electrons captured around such internal defects and dangling bonds, indirectly reveal the chemical environment or structural imperfections related to *V*_In_^−^ and *V*_P_^+^. Furthermore, the intensity of these signals is indicative of the concentration of internal defects and dangling bonds; in other words, a higher ESR signal intensity suggests a higher concentration of internal defects and dangling bonds. The integrated ESR signal intensity was calculated by integrating the ESR signal of the magnetic field between 3400 and 3650 mT, because the internal defects and dangling bonds produced a *g*-factor ranging from 1.909 to 1.939, as shown in Figure 3a. The *g*-factor, also known as the Landé *g*-factor, describes the ratio of the magnetic moment to the angular momentum, which determines the interaction of particles with an external magnetic field. In ESR spectroscopy, the *g*-factor provides insights into the electronic environment of the sample, indicating the presence of unpaired electrons associated with defects [94]. In the context of InP quantum dots, bulk vacancy defects such as negatively charged indium vacancies (*V*_In_^−^) and phosphorus-related dangling bonds (P–X) on the surface can significantly influence the ESR signals. These defects generate characteristic ESR signals, allowing for the identification and quantification of defect concentration. The integrated ESR signal intensity decreased remarkably from 5.54 to 0.896 a.u. when the KI doping concentration increased from 0% to 3%. Subsequently, it increased slightly and became saturated when the KI doping concentration increased from 3% to 7%, as shown in Figure 3b. Thus, the lowest integrated ESR signal intensity was observed at a specific KI doping concentration (i.e., 3% KI).

To comprehend why the integrated ESR signal intensity was minimized at a KI concentration of 3%, it is essential to theoretically analyze the dependence of this signal intensity on the number of internal defects and dangling bonds as a function of the KI doping concentration. During the growth of In_x_P_1−x_ core QDs, In^3+^ and P^3−^ were derived from In(OAc)_3_ and TMS_3_P, respectively, to facilitate QD synthesis. Meanwhile, the dopant KI dissociated into K^+^ and I^−^ with a dissociation energy of 3.4 eV [95]. During synthesis, K^+^ and I^−^ can replace *V*_In_^−^ and *V*_P_^+^ within the core QD bulk and passivate the In- and P-dangling bonds on the surface of the core QDs. The K^+^-*V*_In_^−^ substitutes in the In_x_P_1−x_ core QD bulk would be preferentially produced over I^−^-*V*_P_^+^ substitutes, as the diameter difference between K^+^ and *V*_In_^−^ is smaller than that between I^−^ and *V*_P_^+^. In addition, because the surface ligands PA with carboxylic acid functional groups produce H_2_O in the synthesis solution [65,68,96,97], H_2_O chemically oxidizes the oxyphilic P-dangling bonds of the core QD surface, resulting in InPO_x_ well formation on the surface of the core QDs [4,50,56,65,66,67,68]. The presence of I^−^ in the synthesis solution can passivate P-dangling bonds and produce P-I bonds, prohibiting the formation of InPO_x_ on the core QD surface. Note that the oxyphilicity of P is 0.7, whereas that of I is almost 0; thus, the P-I bonds located on the surface of the In_x_P_1−x_ core QDs are difficult to oxidize [65,67,68]. Because the average diameter of In_x_P_1−x_ core QDs increased almost linearly with an increase in the KI doping concentration, as shown in Figure 2, the number of dangling bonds on the core QD surface would increase with the KI doping concentration, given as 4π(diameter/2)^2^. Thus, the passivation degree of P-I did not linearly increase with an increase in the KI doping concentration, yielding the minimum formation amount of the InPO_x_ on the core surface at a specific KI doping concentration (i.e., 3%), as proven later. The diameter distribution of the In_x_P_1−x_ core QDs was minimized at a specific KI doping concentration (i.e., 3%), indicating that the PLQY was maximized and the FWHM was minimized at the KI doping concentration of 3%. A narrower diameter distribution suggests a more mono-disperse QD population, leading to decreased variation in emission wavelengths, a narrower PL spectrum (FWHM), and an enhanced color purity. Conversely, a broader diameter distribution indicates a poly-disperse population. During the annealing step at 320 °C for the core nucleation and growth process, Ostwald ripening occurs, wherein smaller particles dissolve and redeposit onto larger particles due to their higher solubility [98]. This process not only increases the average particle size but also broadens the size distribution, resulting in a redshift of the PL peak and increased FWHM, ultimately degrading the optical properties of the QDs. The number of K^+^-*V*_In_^−^ substitutions increases with increasing KI doping concentration up to 3%, while the number of K^+^ and I^−^ interstitials also begins to increase at a doping concentration of 3%, so that the diameter distribution is minimized at the KI doping concentration of 3%. As a result, the PLQY peaks and the FWHM minimized at a doping concentration of 3%, since a smaller diameter distribution of the In_x_P_1−x_ core QDs results in a higher PLQY and narrower FWHM [4,98,99,100]. Therefore, KI doping during the synthesis of In_x_P_1−x_ core QD can reduce the amount of *V*_In_^−^ dominantly in the core QD bulk and inhibit surface oxidation (i.e., InPO_x_) of the core QDs, thereby increasing the PLQY and significantly narrowing the FWHM. The maximum positive effects (i.e., an increase in PLQY and a decrease in FWHM) occur at the optimal KI doping concentration of 3%.

### 3.2. Dependence of Photo-Optical Properties (i.e., Absorption, PL Spectra, and Time-Resolved PL Decay Curves) on Dopant (i.e., KI) Concentration for Red-Light-Emitting In_x_P_1−x_/Zn_0.6_Se_0.4_/Zn_0.6_Se_0.1_S_0.3_/Zn_0.5_S_0.5_ Core/Multi-Shell QDs

To evaluate the effects of KI doping on the photo-optical properties of In_x_P_1−x_/Zn_0.6_Se_0.4_/Zn_0.6_Se_0.1_S_0.3_/Zn_0.5_S_0.5_ core/multi-shell QDs, the absorption, photoluminescence spectra, light-emitting wavelength, PLQY, and FWHM were measured with respect to the KI doping concentration for the In_x_P_1−x_ core QDs. The PLQY measurements were conducted using the QE-2100 system, which determines the quantum yield by comparing the number of photons emitted to the number of photons absorbed by the sample. The KI-doped core QDs were sequentially subjected to optimal HF treatment and Zn_0.6_Se_0.4_/Zn_0.6_Se_0.1_S_0.3_/Zn_0.5_S_0.5_ multi-shell growth. Note that optimal HF treatment sequentially proceeds the centrifuge of the KI-doped core QDs, the redispersion of QDs in organic solvent, the injection of HF (i.e., 0.14 mmol), and the heating up of the QD-solution up to 210 °C to grow the multi-shell structure. The first absorption peak was redshifted linearly from 597 to 601 nm when the KI doping concentration increased from 0% to 7%, indicating that the average diameter of the core QDs would increase with the KI doping concentration, being strongly correlated with the dependence of the average diameter of In_x_P_1−x_ core QDs on the KI doping concentration in Figure 2f, as shown in Figure 4a and Appendix A. The wavelength of the red-light emission increased almost linearly from 617 to 631 nm as the KI doping concentration in the core QDs increased from 0% to 7%, indicating that the energy bandgap of the KI-doped core QDs decreased with the increasing KI doping concentration, as shown in Figure 3a,b. Moreover, the red-light emission occurred after illumination from an Xe lamp light having a 450 nm wavelength. Comparing the peak absorption wavelength with the red-light-emitting wavelength revealed that the Stokes shift was enhanced from 21 to 30 nm. When the KI doping concentration in the core QDs increased from 0% to 7%, it was implied that photon energy loss into the lattice atoms was enhanced due to the decreasing energy bandgap of the KI-doped core QDs with increasing KI doping concentration. However, the PLQY is not dependent on the emission wavelength. Instead, the emission wavelength is more significantly influenced by the core diameter of the QDs and the bandgap and thickness of the shelling material, which affects the quantized quantum well. The PLQY, on the other hand, is determined by the efficiency with which the absorbed energy is re-emitted as photons at the band edge, rather than being trapped by defects or lost through non-radiative recombination processes due to vibrations or other factors. As the KI doping concentration in the core increases, the emission wavelength redshifts, primarily due to the increase in particle size, as observed in HR-TEM images. At the optimal KI concentration (3%), a minimization of vacancies and interstitial defects was observed, leading to a more uniform core size distribution. This uniform nucleation and growth resulted in the narrowest FWHM and the synthesis of highly uniform cores. By growing a shell layer on these uniformly prepared cores, growth steps like Ostwald ripening, which can adversely affect optical properties, were minimized. Consequently, the PLQY reached its maximum value (~97%). This high PLQY indicates that the process of photon re-emission from the band-edge states was highly efficient, with minimal losses to non-radiative pathways. In addition, the PLQY increased notably from 74% to 97% when the KI doping concentration increased from 0% to 3%. Subsequently, it decreased significantly from 97% to 77% when the KI doping concentration increased from 3% to 7%. Thus, it was maximized at a KI doping concentration of 3%, i.e., 97%, as shown in Figure 4b. In addition, the FWHM decreased rapidly from 43 to 40 nm when the KI doping concentration increased from 0% to 3%. Subsequently, it increased from 40 to 44 nm when the KI doping concentration increased from 3% to 7%. Thus, it was minimized at a KI doping concentration of 3%, i.e., 40 nm, as shown in Figure 4b. In summary, while the emission wavelength redshift with increasing KI concentration is primarily due to particle size growth, the enhanced PLQY at the optimal doping concentration is a result of minimized defects and uniform core-shell structures that promote efficient radiative recombination.

Furthermore, the exciton lifetime of the In_x_P_1−x_/Zn_0.6_Se_0.4_/Zn_0.6_Se_0.1_S_0.3_/Zn_0.5_S_0.5_ core/multi-shell QDs was measured using TRPL spectroscopy as a function of the KI doping concentration in the In_x_P_1−x_ core QDs, as shown in Figure 4c. Note that a longer exciton lifetime via TRPL implies a higher degree of radiative exciton recombination. In particular, the τ_1_ component is associated with band-edge transition emissions, while the τ_2_ component corresponds to defect-associated emissions; an increase in the value of τ_1_ indicates a higher contribution from band-edge emissions [101], resulting in a higher PLQY [102,103]. The biexponential decay function used for fitting is:$$I\left(t\right)=A_{1}e^{-\frac{1}{\tau_{1}}}+A_{2}e^{-\frac{1}{\tau_{2}}}\;$$
where *I*(*t*) is the PL intensity at time *t*, *A*_1_ and *A*_2_ are the amplitudes, and *τ*_1_ and *τ*_2_ are the lifetimes of the fast and slow components, respectively. The average exciton lifetime (*τ*_avg_) was then calculated using the equation:$$\tau_{avg}=\frac{A_{1}\tau_{1}^{2}+A_{2}\tau_{2}^{2}\;}{A_{1}\tau_{1}+A_{2}\tau_{2}}$$

Moreover, to compare the fractions of band-edge transitions and defect-associated emissions in the total lifetime, we calculated the fractions using the following equations:$$\tau_{1}\%=\frac{A_{1}\tau_{1}}{A_{1}\tau_{1}+A_{2}\tau_{2}}\times 100\;\mathrm{and}\;\tau_{2}\%=\frac{A_{2}\tau_{2}}{A_{1}\tau_{1}+A_{2}\tau_{2}}\times 100$$

The detailed PL decay curve components extracted for each condition can be found in Appendix A. When the KI doping concentration increased from 0% to 7%, the average exciton lifetime (*τ*_avg_) increased from 43 ns to a peak of 49 ns at 3% KI doping, then decreased to 45 ns. The proportion of τ_1_ and τ_2_ components showed that for the optimal 3% KI doping, the *τ*_1_ fraction increased from 49% to 51%, while the *τ*_2_ fraction decreased from 51% to 49%. This indicates an enhancement in band-edge transition emissions and a reduction in defect-associated emissions at the optimal doping concentration. Thus, it was maximized at a KI doping concentration of 3%, i.e., 49 ns, as shown in Figure 4d. As expected, the exciton lifetime was well correlated with the PLQY, i.e., a longer exciton lifetime corresponded to a higher PLQY. For the In_0.53_P_0.47_/Zn_0.6_Se_0.4_/Zn_0.6_Se_0.1_S_0.3_/Zn_0.5_S_0.5_ core/multi-shell QDs, both the PLQY and average exciton lifetime were maximized (i.e., 97% and 49 ns, respectively), and the FWHM was minimized (i.e., 40 nm) at a specific KI doping concentration (i.e., 3%).

It could not be conclusively explained whether the PLQY, exciton lifetime, and FWHM were directly associated with *V*_In_^−^, K^+^ and I^−^ interstitials in the core QDs or P-dangling bonds at the interface between the In_0.53_P_0.47_ core and the Zn_0.6_Se_0.4_ shell, or both of them. Thus, for the core/multi-shell QDs, the crystalline properties were characterized using XRD as a function of the KI doping concentration in the core QDs. This could delineate the presence of P-dangling bonds at the interface between the In_x_P_1−x_ core and the Zn_0.6_Se_0.4_ shell, which will be demonstrated. For the undoped In_x_P_1−x_ core QDs grown with the multi-shell layers, crystalline plane peaks of an unknown crystalline plane, {111}, {220}, and {311} were observed at 2θ = 19.8°, 28.2°, 46.9°, and 55.7° (blue XRD intensity line in Figure 5a), indicating a typical zincblende crystalline structure (JCPDS32-0452) of the core and multi-shell QDs [104]. Surprisingly, according to JCPDS 01-072-0144, an unknown crystalline plane was detected at 2θ = 19.8°, corresponding to indium phosphate dihydrate (InPO_4_(H_2_O)_2_) having an orthorhombic crystalline structure. The existence of InPO_4_(H_2_O)_2_ in the core/multi-shell QDs revealed oxidation on the surface of the In_x_P_1−x_ core QDs, although HF treatment proceeded, followed by multi-shell growth. This result clearly proves that the P-dangling bonds at the interface between the In_x_P_1−x_ core and the Zn_0.6_Se_0.4_ shell were present and were chemically oxidized by H_2_O generated from the palmitic acid surface ligands. Otherwise, for the KI-doped core QDs grown with multi-shell layers, the XRD peak intensity at {210} of InPO_4_(H_2_O)_2_ significantly decreased from 114 to 21 a.u. when the KI doping concentration of the core QDs increased from 0% to 3%. It then considerably increased from 21 to 117 a.u. when the KI doping concentration increased from 3% to 7%, as shown in Figure 5a,b. Thus, the XRD peak intensity at {210} of InPO_4_(H_2_O)_2_ was minimized at a specific KI doping concentration (i.e., 3%) In addition, the XRD signal peak intensities corresponding to {111}, {220}, and {311} remained at ~300, ~150, and ~80 a.u., respectively, when the KI doping concentration in the core QDs increased from 0% to 7%, and they slightly decreased with a further increase in the KI doping concentration, as shown in Figure 5a,b. Thus, a KI doping concentration of >3% during core QD growth slightly degrades the zincblende crystalline properties. This result clearly demonstrates that the chemical oxidation degree of the P-dangling bonds on the In_x_P_1−x_ core QD surface during core growth can be significantly reduced by passivating P-dangling bonds with I^−^ (i.e., forming P-I bonds) up to a KI doping concentration of 3%. Beyond this concentration, the passivation effect was noticeably diminished, as the increase in the number of P-dangling bonds significantly exceeded the increase in their passivation degree (i.e., P-I bonds). Note that the average diameter of the In_x_P_1−x_ core QDs increased linearly and significantly with an increase in the KI doping concentration during core QD growth, as shown in Figure 2f. Therefore, the number of P-dangling bonds on the core QD surface was proportional to 0.75 [the KI doping concentration (C_KI_)]^2^, as shown in Appendix A. Meanwhile, the passivation degree of the P-dangling bonds was linearly proportional to the KI doping concentration.

A comparison of Figure 4b and Figure 5b revealed that the dependence of the XRD signal peak intensity at {210} (i.e., InPO_4_(H_2_O)_2_) on the KI doping concentration was well consistent with the dependence of the PLQY, exciton lifetime, and FWHM on the KI doping concentration of the core QDs. In other words, a smaller amount of InPO_4_(H_2_O)_2_ at the interface of the In_x_P_1−x_ core QD and the Zn_0.6_Se_0.4_ shell led to a higher PLQY, a higher exciton lifetime, and a narrower FWHM. These correlation results evidently prove that KI doping of the core QDs can inhibit the chemical oxidation (i.e., InPO_4_(H_2_O)_2_) on the surface of the core QDs prior to HF treatment, thereby significantly enhancing the PLQY and exciton lifetime and narrowing the FWHM [105,106]. Moreover, comparing Figure 2f with Figure 3b and Figure 4d reveals that the average diameter and distribution of the In_x_P_1−x_ core QDs evidently affected the number of non-radiative recombination, PLQY, and FWHM. The PLQY, along with the exciton lifetime and FWHM, peaked at a specific average diameter of the core QDs (i.e., 4.1 nm) and a minimum core QD diameter distribution of ±0.5 nm, corresponding to the KI doping concentration of 3%. These correlations confirm that the number of internal defects, such as *V*_In_^−^, *V*_P_^+^, K^+^ interstitials, and I^−^ interstitials, in the In_x_P_1−x_ core QDs directly influences the PLQY and FWHM. The optimal reduction of *V*_In_^−^ (K^+^-*V*_In_^−^) occurred at the KI doping concentration of 3%, showing a slightly increased average diameter (i.e., from 7.2 to 7.5 nm) and the minimum diameter distribution (i.e., ±1.6 to ±0.9 nm). Thus, the maximum PLQY (i.e., 97%) and a minimum FWHM (i.e., 40 nm) were observed at the optimal KI doping concentration (i.e., 3%). However, over-doping of KI (i.e., >3%) increased the number of K^+^ and I^−^ interstitials in the In_x_P_1−x_ core QDs; thus, both the average diameter and diameter distribution of the core QD increased significantly, leading to an increase in the number of P-dangling bonds on the core QD surface. As a result, the PLQY decreased and the FWHM increased significantly beyond a KI doping concentration of 3%. Therefore, the above correlation clearly demonstrates that PLQY and FWHM were principally and preferentially determined by both the reduction of *V*_In_^−^ via the formation of K^+^-*V*_In_^−^ substitutes in the In_x_P_1−x_ core QD bulk and the inhibition of the chemical oxidation of P-dangling bonds (i.e., InPO_4_(H_2_O)_2_) via the formation of P-I bonds.

### 3.3. Color Gamut Performance of QD Functional CF-OLED Hybrid Display Using KI-Doped R-, G-, and B-QD Functional CFs and Blue OLED BLU

The application of red-light-emitting KI-doped In_0.53_P_0.47_/Zn_0.6_Se_0.4_/Zn_0.6_Se_0.1_S_0.3_ /Zn_0.5_S_0.5_ QDs, which were carefully designed for precisely tuned optical properties (i.e., narrow FWHM of ~40 nm and high PLQY of ~97%) through the KI-doping process, demonstrated a significant improvement in color representation. The color representation performance of the QDCF-OLED hybrid displays was assessed to evaluate their potential as Ultra-High-Definition (UHD) displays by comparing their color gamut coverage to the NTSC and Rec.2020 standards [85], as shown in Figure 6.

The structure of the QDCF-OLED hybrid display consisted of an emitting layer (QD functional CF) on the top and a backlight unit (blue-OLED BLU) on the bottom, as shown in Figure 6a. The top emitting layer, which was called a QD functional CF, was patterned with a mixture of optimized red-light-emitting In_0.53_P_0.47_/Zn_0.6_Se_0.4_/Zn_0.6_Se_0.1_S_0.3_/Zn_0.5_S_0.5_ core/multi-shell QDs doped with 3% KI, green-light-emitting InP/ZnSe/ZnSeS/ZnS core-multi-shell QDs and blue-light-emitting ZnSe/ZnS core–shell QDs with conventional R-, G-, and B-color filters. The B-, G-, and R-CFs coated on quartz glass used in this study exhibited broad transmittance spectra at 371–563 nm for blue, 478–595 nm for green, and >570 nm for red, with peak transmission spectra at 451 nm for blue, 527 nm for green, and >631 nm for red, as shown in Figure 6b. In addition, the optical properties of the InP-based G-QDs and ZnSe-based B-QDs were characterized using PL spectroscopy. The blue and green peak wavelengths were observed at 449 and 538 nm, with FWHMs of 32 and 36 nm and absolute PLQYs of 71% and 94%, respectively, as shown in Appendix A. The bottom layer of the BLU contained a blue light-emitting OLED with a PL spectrum that peaked at 446 nm and an FWHM of 96 nm. The properties of the blue OLED device are detailed in Figure 4b and the Appendix A (see Appendix A).

The light-emitting mechanism of the B-, G-, and 3% KI-doped R-QDs functional CFs was associated with energy-down-shift (EDS), which absorbed the blue-light energy from the blue-OLED BLU and emitted R-, G-, and B-light through the R-, G-, and B-QDCFs, respectively [107,108,109]. The PL peak wavelengths and FWHMs for R-, G-, and B-emitting light were 635 and 28 nm, 533 and 22 nm, and 447 and 29 nm, respectively, as shown in Figure 6c and Table 1. The QDCF-OLED hybrid display clearly demonstrated the absence of crosstalk between the B- and G-light emissions, as well as between the G- and R-light emissions. In particular, the QD functional-CF OLED employing 3% KI-doped R-QDs presented a significantly narrower FWHM of 28 nm. The FWHM of red-light emission via the QD functional CF OLED for 3% KI-doped R-QDs (i.e., 28 nm) was reduced by ~6 nm compared with that for undoped R-QDs (i.e., 34 nm), as shown in Table 1. In addition, to assess the color-representation performance of high-resolution displays, the color gamut performance of the R-, G-, and B-QD-functionalized CF using a blue OLED BLU was estimated. By comparing the color gamut performance of devices fabricated with undoped core/multi-shell QDs and those doped with 3% KI, the enhanced color representation achieved through doping was clearly demonstrated. The color gamut for the QD functional-CF OLED, covering the NTSC and Rec.2020 standards [86,87,89], for devices using undoped R-QDs and 3% KI-doped QDs was 125.8%, 94.2%, 131.1%, and 98.1%, respectively, as shown in Figure 6d and Table 1. The color gamut for the QD functional-CF OLED using 3% KI-doped R-QDs showed improvements of 5.3% and 4% for the NTSC and Rec.2020 standards, respectively, compared with those using undoped R-QD functional-CF OLEDs. This result, compared with recent research results summarized in Table 1, indicates that significantly reducing the FWHM of KI-doped In_0.53_P_0.47_ core QDs grown with multiple shells for red-light emission can substantially enhance the color gamut (i.e., NTSC 131.1%, Rec. 2020 98.2%) for QD-functional CF-OLED hybrid displays using KI-doped R-, G-, and B-QD functional CFs and blue OLED BLU and demonstrated improved color gamut coverage.

## 4. Conclusions

For recently developed QD-OLED displays for TVs and monitors, the PLQY and FWHM of In_0.53_P_0.47_ core QDs, grown with Zn_0.6_Se_0.4_/Zn_0.6_Se_0.1_S_0.3_/Zn_0.5_S_0.5_ multi-shells emitting red light, have been essentially maximized and minimized, respectively. Internal defects in the core QD bulk (i.e., vacancies and interstitials) and dangling bonds at the core-ZnSe shell interface predominantly reduce the PLQY and broaden the FWHM.

In our study, KI doping during the growth of In_0.53_P_0.47_ core QDs significantly reduced the concentration of vacancies (i.e., *V*_In_^−^) within the core QD bulk. This reduction in vacancies was evidenced by a decreased relative intensity of non-radiative unpaired electrons in the core QDs. Consequently, the average diameter of the In_0.53_P_0.47_ core QDs increased by 0.4 nm, and the diameter distribution narrowed by ±0.5 nm compared to undoped In_0.56_P_0.44_ core QDs due to the substitution of *V*_In_^−^ with K^+^ from KI doping during synthesis. The optimal KI doping concentration was found to be 3%, beyond which interstitial defects (i.e., K^+^ and I^−^) were generated, reducing PLQY and increasing FWHM due to an increase in non-radiative unpaired electrons and dangling bonds. In addition to reducing vacancies, KI doping effectively inhibited oxidation on the core QD surface during synthesis. P-dangling bonds on the surface of the core QDs were chemically oxidized by H_2_O from surface ligands, forming InPO_4_(H_2_O)_2_, identified by the presence of {210} in core QDs grown with multi-shells, which significantly reduced PLQY and increased FWHM. The dissociated I^−^ from doped KI passivated the P-dangling bonds on the QD surface by forming P-I bonds, significantly reducing InPO_4_(H_2_O)_2_ formation and enhancing PLQY. Optimal KI doping concentration (3%) maximally inhibited InPO_4_(H_2_O)_2_ formation at the In_0.53_P_0.47_ core QD and Zn_0.6_Se_0.4_ shell interface. Over-KI doping (>3%) led to an increase in the average core QD diameter, which, in turn, increased the number of P-dangling bonds, ultimately resulting in higher InPO_4_(H_2_O)_2_ formation.

Thus, the KI-doped In_0.53_P_0.47_/Zn_0.6_Se_0.4_/Zn_0.6_Se_0.1_S_0.3_/Zn_0.5_S_0.5_ QDs emitting red light exhibited a PLQY of ~97% and an FWHM of ~40 nm, indicating their suitability for QD-OLED displays without R-, G-, or B-crosstalk, achieving a near-perfect color gamut (i.e., ~131.1% @ NTSC and ~98.2 @ Rec.2020).

## Figures and Tables

**Figure 1 nanomaterials-14-01055-f001:**
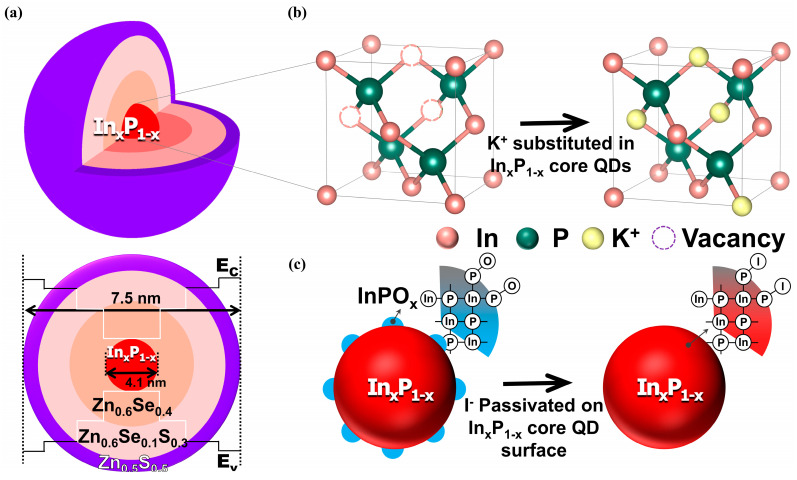
Effects of the KI doping on the In_x_P_1−x_ core QDs. (**a**) Structure of red-light-emitting In_x_P_1−x_/Zn_0.6_Se_0.4_/Zn_0.6_Se_0.1_S_0.3_/Zn_0.5_S_0.5_ core/multi-shell QDs, (**b**) substitution of vacancy defects in the zincblende crystalline structure of the In_x_P_1−x_ core QDs, and (**c**) passivation of P-dangling bonds on the In_x_P_1−x_ core QD surface.

**Figure 2 nanomaterials-14-01055-f002:**
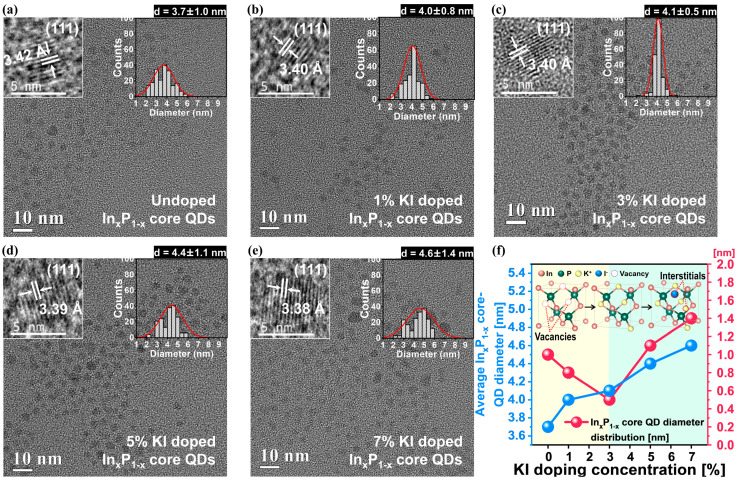
In_x_P_1−x_ core QDs’ average diameter and diameter distribution depending on the KI doping concentration. HR-TEM images of the core QDs with different KI doping concentrations: (**a**) undoped, (**b**) 1% KI, (**c**) 3% KI, (**d**) 5% KI, (**e**) 7% KI, and (**f**) Average In_x_P_1−x_ core QD diameter and diameter distribution for various KI doping concentrations.

**Figure 3 nanomaterials-14-01055-f003:**
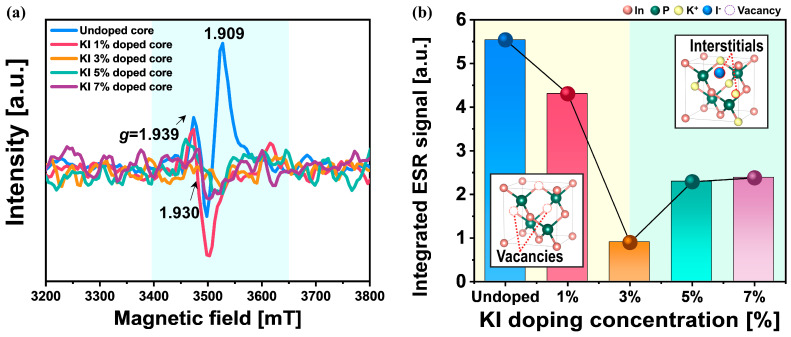
Relative unpaired electron concentration of the In_x_P_1−x_ core QDs depending on the KI doping concentration. (**a**) ESR spectra with the *g*-factor of KI-doped core QDs and (**b**) relative integrated ESR signal under a magnetic field between 3400 and 3650 mT.

**Figure 4 nanomaterials-14-01055-f004:**
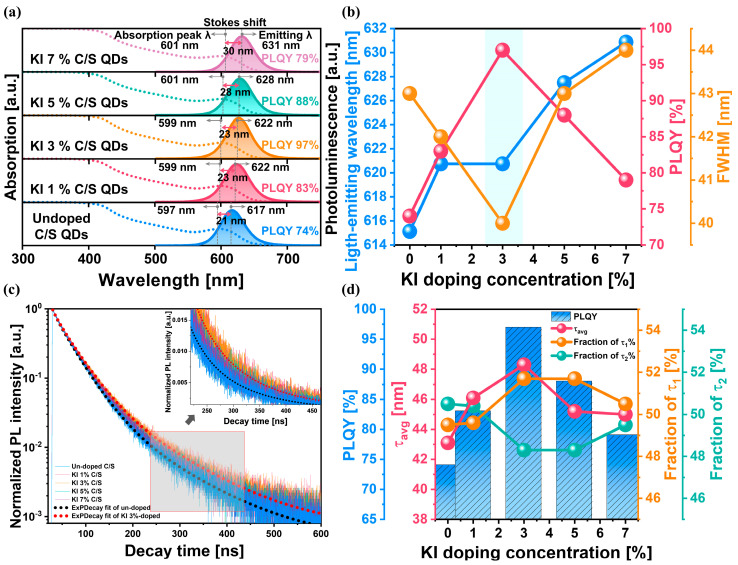
Optical properties of KI-doped red-light-emitting In_x_P_1−x_/Zn_0.6_Se_0.4_/Zn_0.6_Se_0.1_S_0.3_/Zn_0.5_S_0.5_ core/multi-shell QDs depending on the KI doping concentration. (**a**) Absorption and PL spectra and (**b**) light-emitting wavelength, PLQY, and FWHM. (**c**) Normalized time-resolved PL decay curves and (**d**) relative exciton lifetime includes the calculated average lifetime (τ_avg_) for each KI doping concentration and the fraction of lifetime components (τ_1_% and τ_2_%).

**Figure 5 nanomaterials-14-01055-f005:**
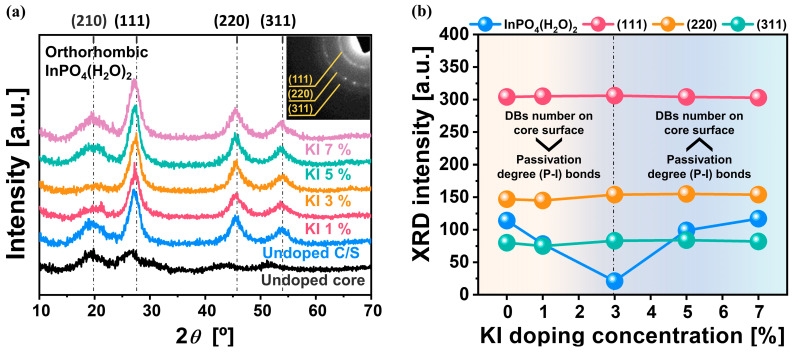
Crystallinity of InP-based core and core/multi-shell QDs depending on KI doping concentration. (**a**) XRD patterns of KI-doped red-light-emitting In_x_P_1−x_/Zn_0.6_Se_0.4_/Zn_0.6_Se_0.1_S_0.3_/Zn_0.5_S_0.5_ core/multi-shell QDs, and (**b**) relative XRD peak intensities on the crystalline planes.

**Figure 6 nanomaterials-14-01055-f006:**
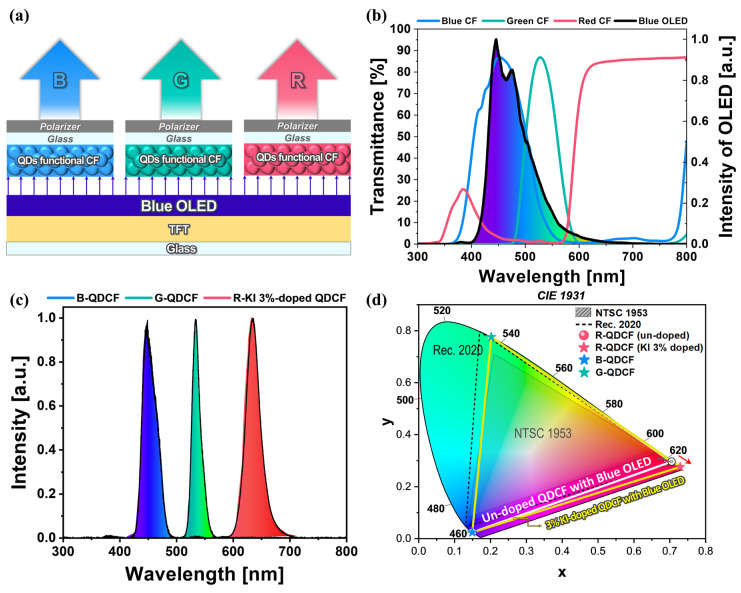
Color gamut properties of the QD-functional CF OLED. (**a**) Schematics of the QDCF OLED; (**b**) PL spectra of the blue OLED and transmittance spectra of B-, G-, and R-CFs; (**c**) PL spectra of QD-functional CF OLED using B-, G-, and KI-doped R-QDCF and blue OLED BLU; and (**d**) comparison of the RGB primary color triangles between the QD-functional CF OLED and KI-doped R-QDCF OLED under the blue OLED in the CIE 1931 color space.

**Table 1 nanomaterials-14-01055-t001:** Optical characteristics, CIE coordinates, and color representation of QD Functional CF OLED.

Applications	Light-Emitting Color	WL [nm]	FWHM [nm]	CIE Coordinates [x, y]	Color Gamut		Ref.
					NTSC	Rec. 2020 (BT.2020)	
QD functional-CF OLED for undoped R-QD	Blue	447	29	(0.15, 0.02)	125.8%	94.2%	Our work
	Green	533	22	(0.20, 0.77)			
	Red	628	34	(0.70, 0.29)			
QD functional-CF OLED for 3% KI doped R-QD	Blue	447	29	(0.15, 0.02)	131.1%	98.2%	
	Green	533	22	(0.20, 0.77)			
	Red	635	28	(0.72, 0.27)			
Flexible full-color active-matrix QD-OLED display 13.6 inch (1920 × 1080)	Blue	461	26	(0.137, 0.063)	93.0%	-	[110]
Inkjet-printed 6.6-inch red active matrix QD-OLED display panel with a resolution of 384 × 300 pixels with R-CdSe/ZnS QDs	BOLED	462	21	(0.136, 0.0693)	The maximum light conversion efficiency reached 32.7%		[111]
	QD-BOLED	630	35	(0.721, 0.279)			

## Data Availability

Data are contained within the article and Appendix A.

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
