# Peer review of "Potassium Iodide Doping for Vacancy Substitution and Dangling Bond Repair in InP Core-Shell Quantum Dots"

_nanomaterials, 2024, doi:10.3390/nano14121055_

Round 1

Reviewer 1 Report

Comments and Suggestions for Authors

In this work, the authors proposed the potassium iodide (KI) doping during the synthesis of the In0.53P0.47 core quantum dots (QDs), which can eliminate In- vacancies in the core QD bulk by forming K+-VIn- substitutes and inhibit the formation of InPO4(H2O)2 at the core QD–Zn0.6Se0.4 shell interface through the passivation of phosphorus (P)-dangling bonds by P-I bonds. The photoluminescence quantum yield (PLQY) of ~97% and full width at half maximum (FWHM) of ~40 nm were achieved by modifying the KI doping concentration. This work will be of great interest to researchers in the field. I would like to give some comments and suggestions. The detailed comments are as follows:

1) In introduction, the authors write: “…Among various luminescent materials, quantum dots (QDs) have been intensively researched owing to their simple wavelength tunability, high efficiency, high stability, and high color purity [1-10]. Consequently, in recent decades, research on the application of QDs in fields such as solar energy [11-14], bioimaging [15,16], and displays [10,17-21] has been vigorously pursued. In particular, luminescent materials for display applications, such as quantum-dot organic light-emitting diodes (QD-OLED) and quantum-dot light-emitting diodes (QLED), are required to achieve a photoluminescence quantum yield (PLQY) of >95% and a full width at half maximum (FWHM) of <30 nm [1,10,22-24].…” The general reference list in the introduction seems a bit thin, considering the evolution in the field within the recent years. The combination of QDs with GaN-based LEDs has been used for the realization of full-color displays. To give the readers a much broader view, recent developments concerning on this topic, such as Laser & Photonics Reviews 2023, 17, 2200455; Optics Letters 47(5), 1291-1294 (2022), etc. should be added, so that the readers can be clear about the state-of-the-art of this topic.

2) Please provide the definition and/or calculation equation for PLQY.

3) In paragraph 261, the author write:The diameter distribution of the InxP1-x core QDs was minimized at a specific KI doping concentration (i.e., 3%), indicating that the PLQY was maximized and the FWHM was minimized at the KI doping concentration of 3%.Please explain the correlation between the diameter distribution of core QDs and the PLQY and FWHM.

4) In paragraph 231, the author write:because the internal defects and dangling bonds produced a g-factor ranging from 1.909 to 1.939, as shown in Figure 3a.Please provide the physical meaning of the g-factor and add some comments regarding the correlation between the g-factor and defects.

5) In figure 4(a), the PL peak was redshifted as the KI doping concentration increases. Could the author believe that the PLQY was wavelength-dependent? In other words, the discrepancies in PL peak for QDs may also affect the PLQY and FWHM.

6) What level does the OLED produced by this method attain within the reported researches? It is better to add comparisons with existing report.

Reviewer 2 Report

Comments and Suggestions for Authors

The manuscript presents a comprehensive study of doping InP QDs with KI and its implications on structural and optical properties of the core particles as well as on the final core/multishell QDs. The study is solid and provides sufficient experimental evidence of the authors' claims. Purely as a recommendation, I would suggest starting discussion of results with optical properties of the QDs, as it is the main focus of the whole work, and continuing with their structural characterisation (i.e. TEM, XRD, ESR), which give a deeper insight into the QD structure and explain changes in optics. This is especially relevant, as the authors start discussing optical properties on page 7, before their introduction. Among minor corrections are the following.

1. The title of the work is too long and can be shortened for a better comprehension by the readers.

2. "A halide metal", which appears in the introduction, should be metal halide.

3. In the materials list, the authors use abbreviation Zn(st)2, while in the synthesis description Zn(SA)2 appears. This should be corrected.

4. Page 4, line 131: "the distilled mixed solution" should be degassed, if I understand it correctly.

5. Page 5, line 205-206, page 6, line 245-246: unnecessary repetition of the same sentence.

6. First, in the main text Figure S5 is mentioned on page 4, i.e. the numbering is not consistent (should start with S1, etc.).

7. In the supp. info. in Figure S6c the PL spectrum does not correspond to the scale (shifted to shorter wavelengths).

Comments on the Quality of English Language

1. Page 4, line 172-173: "for the application of QD-functional CF-OLED hybrid display applications" should be "for the application in QD-functional CF-OLED hybrid displays"

2. Page 5, line 206: "When the KI doping from ±0.9 to ±0.4 nm, as shown in Figures 2a-c and f." - incomplete sentence.

3. Page 8, line 285: "to grow the multi-shell growth" - tautology.

Reviewer 3 Report

Comments and Suggestions for Authors

The manuscript ‘Bulk Vacancy Substitution and Interface Phosphorus-Dangling Bond Repair via Potassium Iodide Doping during Core Synthesis in Indium Phosphide Core-Shell Quantum Dots’ authored by Lee et al. synthesized InxP1-x core QDs that are grown with Zn0.6Se0.4/Zn0.6Se0.1S0.3/Zn0.5Se0.5 multi-shells while doping KI to reduce the concentration of vacancies (i.e., VIn-) within the bulk of the core QD. The authors found that the highest PLQY of 97% in red emitting light is achieved at a KI doping concentration of 3%, contributing to the performance enhancement of QD-OLED displays. This work is well designed, and the manuscript is written in a good manner. However, I cannot recommend accepting this manuscript in Nanomaterials before the authors address the concerns below.

Major comments:

1.     The title is a bit lengthy. Please compress it with the most important information.

2.    In Introduction, there is lack of literature review about diverse doping methods for this type of InP QDs.

3.  The actual KI doping amounts should be clarified by conducting elemental analysis such as ICP-MS and EDS.

4.     The XPS measurements are mandatory for determining the surface oxidized state of InP core QDs.

5.   TRPL decays in Figure 4c do not make much sense by considering the big difference that is more than 20% in PLQYs of undoped and 3% doped QDs. Basically, there is no obvious change in lifetime before the PL intensity drops to 10-2 level (about 0.01% of the PL peak intensity). The authors should show the function that fitted the decays and clearly assign the components to specific factors. How did the authors extract the lifetime? The TRPL data required recheck and reanalysis in depth.

6.    What does ‘NTSC’ mean? For all abbreviations, please describe it in full name when first time to mention.

7.   There are some typos and errors that require to be rectified, e.g., Line 439 Zn0.5Se0.5 should be Zn0.5S0.5.

Comments on the Quality of English Language

 Minor editing of English language is required.

Reviewer 4 Report

Comments and Suggestions for Authors

In this work, the authors suggested that KI doping during the growth of the In0.53P0.47 core QDs significantly reduced the concentration of vacancies (i.e., VIn-) within the bulk of the core QD. The result is good and I recommend it for publication after minor revision. Comments:

1. The abstract should be refined to highlight the novelty, principle and importance of this work, but not the detailed results.

2. The conclusion should be shortened. Some of the statements were copied from those in Introduction.

Reviewer 5 Report

Comments and Suggestions for Authors

The paper nanomaterials-3024309 deals with the synthesis, the study of the optical properties, and the application in Q-LED of Indium Phosphide Core-Shell Quantum Dots. The general topic of the paper follows the aims and the scope of the Nanomaterials Journal and it could be interesting for a relatively broad audience. However, the paper in its present form needs some improvements.

1) Figure 1 reports a scheme of the core-shell structure with precise dimensions that are not discussed and proved in the rest of the manuscript. It is not clear if it is a structure that the authors aim to obtain or if represents what they are going to show in the rest of the paper.

2) Figure 2 The TEM measurements reported are far to be clear but mostly it is not clear how you correlate the results reported in the inset with the values of the shell reported in panel f. Further, the error in the measurements indicates that all the values can fall in the same range.

3) Page 8 - All the values reported for the optical measurements give an associated error <1 nm. Are you sure that the resolution of the measure performed is below 1 nm? 

4) The experimental TRPL reported in Figure 4 is very noisy and the difference between 35.90 and 38.95 ns cannot be really appreciated. It seems just a possible analytical result. I suppose that the error associated to the fitting procedure is well above a few ns. I strongly suggest repeating the measurements in a shorter time window to appreciate the fast time behavior and to further increase the time window to verify the presence of multiple decays.

Figure 5 panel b is not clear. The TEM image of the background does not seem very useful, but mainly it is not very readable

As a general consideration, the paper reports a detailed discussion, sometimes difficult to follow with respect to the experimental values reported.   

Comments on the Quality of English Language

Difficult to follow. The authors need to edit the paper to increase the readability 

Round 2

Reviewer 3 Report

Comments and Suggestions for Authors

The authors have effectively addressed my previous concerns and I recommend accepting this manuscript in Nanomaterials.

Comments on the Quality of English Language

Minor editing of English language is required.

Author Response

Dear Reviewer,

Thank you very much for taking the time to review our manuscript and for your thoughtful comments. Your suggestions have significantly improved the quality of our work.

Reviewer 5 Report

Comments and Suggestions for Authors

The paper nanomaterials-3024309 still lacks of clarity in the measure reported.

Defining the resolution of the measure you cannot indicate 12 digital places. This is an electronic precision that is not the resolution of the instrument. As you can see from the webpage of the product (https://otsukael.co.kr/qe2100)theresolution depend on the model that you utilized and it is clearly indicated in nm. A part this number you have to define the error on your specific measure and, in case of relatively broad emission (as in your case) 0.1 nm as precision is difficult to appreciate.

About the time resolved measurements. The results indicated is just a fitting procedure that give an error that I suppose well above 0.01 ns... 

About the dimension reported in figure 2 a-e (inset) and summarized in panel f. I do not see any variation. several values are within the error of the measures reported (for example you can easily affirm that all the average diameter are constant at about 4 nm) 

Comments on the Quality of English Language

Nothing to declare

Round 3

Reviewer 5 Report

Comments and Suggestions for Authors

The authors corrected adequately the paper.

Comments on the Quality of English Language

Ok